# Bioactive Compounds and Biological Activities of Sorghum Grains

**DOI:** 10.3390/foods10112868

**Published:** 2021-11-19

**Authors:** Zhenhua Li, Xiaoyan Zhao, Xiaowei Zhang, Hongkai Liu

**Affiliations:** 1College of Agriculture, Guizhou University, Huaxi District, Guiyang 550025, China; 2Department of Food Science and Nutrition, College of Culture and Tourism, University of Jinan, No. 13 Shungeng Rd., Jinan 250002, China; st_zhaoxy@ujn.edu.cn (X.Z.); st_zhangxw@ujn.edu.cn (X.Z.)

**Keywords:** sorghum, bioactive compounds, phenolic compounds, biological activities

## Abstract

Sorghum is the fifth most commonly used cereal worldwide and is a rich source of many bioactive compounds. We summarized phenolic compounds and carotenoids, vitamin E, amines, and phytosterols in sorghum grains. Recently, with the development of detection technology, new bioactive compounds such as formononetin, glycitein, and ononin have been detected. In addition, multiple in vitro and in vivo studies have shown that sorghum grains have extensive bio-logical activities, such as antioxidative, anticancer, antidiabetic, antiinflammatory, and antiobesity properties. Finally, with the establishment of sorghum phenolic compounds database, the bound phenolics and their biological activities and the mechanisms of biological activities of sorghum bioactive compounds using clinical trials may be researched.

## 1. Introduction

In the last twenty years, frequent viral outbreaks, such as the recent COVID-19 outbreak that has caused massive numbers of deaths around the world, have been highly contagious and easily transmissible [1,2]. Therefore, safe and effective interventions are urgently needed to prevent, reduce susceptibility, and lessen all kinds of viruses [3]. The consumption of nutraceuticals, functional foods, or herbal plants could help to prevent and manage viral infections [1,4]. Additionally, preventive effects could be related to the presence of several bioactive compounds (or natural products) in nutraceuticals, functional foods, and herbal plants [1,5]. Numerous studies have shown that bioactive compounds have various kinds of biological activities, such as antioxidant, anti-inflammatory, and antimicrobial properties, which help to protect against human disease [6].

Sorghum (*Sorghum bicolor* L. Moench) is a dietary staple in the Americas, Asia, Australia, and Africa, and it is the fifth most cultivated cereal in the world [7,8,9]. It is gluten-free and drought-tolerant among major cereal grains [10,11]. In particular, it is unique compared to other major cereal grains for having various bioactive compounds such as phenolic acids, procyanidins, flavonoids, and anthocyanins [8,12,13]. Additionally sorghum is the only dietary source of 3-deoxyanthocyanidins (3-DXAs) and even contains the highest amount of phenolic compounds among cereal grains [14]. Multiple studies have shown that bioactive compounds in sorghum grains can benefit the gut microbiota and have extensive biological activities, such as anti-inflammation, antioxidation, antithrombotic, and antidiabetic properties [8,15,16]. 

Currently, there are several informative reviews depicting the bioactive compounds and biological activities in sorghum grains [17,18,19,20], whereas there are few studies on the factors influencing sorghum bioactive compoundsand new bioactive compounds being found in sorghum grains nearly two years ago. Hence, the present review aims to summarize the data related to bioactive compounds and biological activities in sorghum grains and analyze the influencing factors or mechanism.

## 2. Bioactive Compounds in Sorghum Grains

Bioactive compounds are widely distributed in plant source foods and most are secondary metabolites. Sorghum grain is a good source of bioactive compounds. Here, we summarize the phenolic compounds (Table 1 and Table 2) and carotenoids, vitamin E, amines, and phytosterols (Table 3) in sorghum grains.

### 2.1. Phenolic Compounds

Phenolic compounds are important secondary metabolites with significant physiological benefits for humans [47]. They contain at least one aromatic ring and one or more hydroxyl groups in their chemical structures and range from simple phenolic acids to highly polymerized tannins [46,47,48]. A wide class of phenolic compounds has been found in sorghum, including phenolic acids, flavonoids, stilbenoids, and tannins (Table 2).

Table 1 shows total phenolic compounds (TPC) in sorghum grains. The presence of TPC in most sorghum whole grain is 0.46 ~ 20 mg GAE/g (Table 1). Additionally, the highest content of TPC has been reported in a red sorghum whole grain, totaling 47.86 mg GAE/g [24]. The content of TPC reported in sorghum bran varies even more, ranging from 0.18~70 mg GAE/g (Table 1). The difference in the content of TPC in sorghum depends on many factors.

The characteristic of the variety is an important factor in determining the content of TPC in sorghum. A significant difference in total phenolic content has been observed between five different sorghum varieties with different seed coat color; the content of TPC in red pericarp sorghum, brown pericarp, black pericarp, pearl white pericarp, and white pericarp sorghum was 1040.73 ± 6.79, 955.88 ± 9.91, 844.21 ± 8.92, 191.18 ± 3.87, and 173.68 ± 3.11 mg GAE/100 g, respectively [10]. Awika et al. (2005) observed that brown sorghum grains possessed higher total phenolic content compared to the black pericarp and white pericarp [25]. Burdette et al. (2010) reported that TPC contents of sumac (red) and black sorghum bran varieties were 20- and 7.5-fold greater than that of white sorghum bran extract and 8.9- and 3.3-fold greater than that of Mycogen (bronze) sorghum bran extract, respectively. Furthermore, most black sorghum bran possessed higher total phenolic compound content as compared to the brown pericarp bran, and the content of TPC in black sorghum bran showed a huge difference [26]. Therefore, the color of pericarp is not an ideal marker of TPC [10].

The extraction method, the first stage affecting the research and utilization of phenolic compounds, is another important factor in determining the content of TPC in sorghum. As seen in Table 1, solid–liquid extraction is a common method used to extract phenolic compounds. Some of the most widely used solvents in the extraction of phenolic compounds include methanol, ethanol, and acetone (Table 1). The extraction solvent (methanol, ethanol, and acetone), solvent concentration (40%, 60%, and 80%, *v/v*), and solvent-to-solid ratio (10:1, 20:1, 30:1, and 40:1, mL/g) on the extraction yields of TPC from the defatted red sorghum were evaluated; results showed that the optimized extraction conditions involved the red sorghum being extracted with acetone/water mixture (60:40, *v/v*) at the solvent-to-solid ratio of 30:1 [8]. Many solvent factors can influence the extraction efficiency of phenolics compounds, and even different studies may have different results. TPC in ethanolic extracts has been detected in higher concentrations than in methanolic [16]; the water extract showed the highest TPC, followed by 60% t-butanol, 60% ethanol, and 60% methanol [22]. Ethanol extracts contained higher concentrations of TPC than its aqueous counterparts [23]. Transforming phenolics compounds from solid matrices to extraction solvent depends on solvent polarities and the physicochemical properties of compounds [23]. Hence, it is important to choose a proper solvent or solvent composition for targeting active compounds. Moreover, interfering with the substances in the extraction solvent, such as HCl and formic acid, can influence the extraction efficiency of phenolic compounds [49]. Acidic environments can stimulate the release of the bound phenolic compounds and the hydrolysis of flavonoid glycosides [50,51]. In addition, some emerging technologies, such as subcritical water extraction and ultrasound-assisted extraction, have been used to extract phenolic compounds from sorghum [8,28]. Some emerging technologies, including enzymatic, pulsed-electric field, accelerated solvent, supercritical fluid, and microwave treatment, that are used to extract phenolic compounds from sorghum have not been reported. In view of the fact that traditional extraction methods consume a lot of solvent, a more environmentally friendly extraction method for the extraction of sorghum phenolics needs to be developed.

However, most studies have so far focused on the identification and biological activity of free phenolic compounds in sorghum. While phenolic compounds in sorghum are present both in free and bound forms, most phenolic compounds in sorghums exist in bound form [33]. Bound phenolic compounds link to structural components of the cell wall and hamper phenolic compounds’ bioaccessibility and bioavailability [33,52]. Therefore, it is important to seek out ways to promote the release of bound phenolic compounds and increase the phenolic compounds’ bioaccessibility and bioavailability.

#### 2.1.1. Phenolic Acids

Numerous phenolic acids had been found in native and processed sorghum grains (Table 2). In recent studies, phenolic acids in sorghum were identified by high performance liquid chromatography (HPLC) on the basis of previous studies. Additionally, the number of phenolic acids identified in sorghum has varied from study to study. Caffeic acid, p-coumaric acid, sinapic acid, gallic acid, protocatechuic acid, p-hydroxybenzoic acid, and ferulic acid have been studied more in the above phenolic acids, and ferulic acid has been the predominant phenolic acid. For example, ferulic, p-coumaric, caffeic, and 3,4-dihydroxybenzoic acids were identified in a red sorghum; ferulic acid was the predominant phenolic acid [33]. Moreover, ferulic, p-coumaric, and protocatechuic acids had the highest concentrations among the 11 assayed phenolic acids in both red and white sorghum grain [36], and ferulic acid was the most prominent phenolic acid and was higher in red and brown sorghum [10]. In addition, the contents of individual phenolic acids may be significantly different among different sorghum genotypes [10,12,33]. Moreover, the contents of most bound phenolic acids are higher than those of the corresponding soluble forms [33]. Therefore, it is necessary to release bound phenolic acids to their soluble forms from the sorghum matrix by using various treatment techniques.

#### 2.1.2. Flavonoids 

Many flavonoids have been found in sorghum grains (Table 2). Sorghum is the only dietary source for 3-DXAs. Luteolinidin (LUT), apigeninidin (AP), 5-methoxyluteolinidin, and 7-methoxy apigeninidin are four major forms of 3-DXAs [10,12,33,53]. 3-DXAs primarily exist in plant tissue as aglycones [33]. The sorghum genotype significantly affects the content and composition of 3-DXAs in sorghum grain. M. Li et al. (2021) reported that LUT was the predominant 3-DXA, with its total content accounting for 40.55 to 78.36% of the total 3-DXAs in their study [33]. LUT and AP were higher in red and brown sorghum grains followed by black in comparison to white pericarp sorghum varieties [10]. The difference in 3-DXAs between sorghum genotypes may be attributed to the difference in chalcone synthase and flavonoid-3′-hydroxylase, which are involved in the biosynthesis of 3-DXAs. Moreover, 3-DXAs are present in free forms and stable in solution compared to other anthocyanidins [33,54]. Hence, 3-DXAs are mostly water-soluble pigments in sorghums.

Among flavones in sorghum grains, the most well-known compounds are luteolin and apigenin, and naringeninis is the most well-known compound in flavanones. Additionally, among the class of flavonols, kaempferol and quercetin are the most investigated, and catechin is the most investigated in the flavanols of sorghum grains. Taxifolin is the most investigated in the dihydroflavonols of sorghum grains [28] (Table 2). 

Anthocyanins have two double bonds and a hydroxyl group at C3 [55]. Ofosu et al. (2021) showed in their study that cyanidin was identified in all three sorghum genotypes [12]. However, there are relatively few reports on the research of anthocyanins in sorghum grains. 

Isoflavones are the only flavonoids that have the benzene ring at C3 [55]. They are naturally synthesized in legumes, however, formononetin, glycitein, and ononin were reported for the first time in sorghum by Ofosu et al. (2021) [12]. Hence, the content of isoflavones in other sorghum varieties needs to be verified in further study.

#### 2.1.3. Stilbenoids 

Stilbenoids are a class of substances with a stilbene parent core and a polymer. Sorghum has the capability of producing stilbenoids metabolites [56]. Research has shown that 0.4–1 mg/kg amount of *trans*-piceid and up to 0.2 mg/kg trans-resveratrol were quantified in red sorghum grains [38]. While few studies are relevant to stilbenoids in sorghum grain, the metabolic regulation and variety of difference in stilbenoids in sorghum grains need to be studied and explained.

#### 2.1.4. Tannins

Based on structural characteristics, tannins can be classified into hydrolysable tannins and condensed tannins (proanthocyanidins) [55]. Proanthocyanidins are unique in some cereal grains, however there are comparatively more reports about proanthocyanidins in various sorghum varieties. Perhaps the contents of proanthocyanidins in sorghum are enough to yield astringency and a bitter taste due to their complexation and precipitation of proteins. Hence, tannins are considered as anti-nutrients yet have attracted more attention due to increasing knowledge of their health benefits. 

### 2.2. Carotenoids

Carotenoids are C40 isoprenoids and have many beneficial effects on human health [57]. Three carotenoids, lutein, zeaxanthin, and β-caroteneis, are the most investigated in sorghum grains (Table 3), and the main sorghum carotenoids are xanthophylls (lutein and zeaxanthin) [40]. The contents of carotenoids have varied in various studies [36,39,40]. This may be due to the difference in genotypes, extraction methods, detection methods, and sorghum grain fractions. For example, the total carotenoid content varied from 2.12 to 85.46 μg/100 g in one hundred sorghum genotypes [40]. The high variability in the content of carotenoids in sorghum grains is due to the expression status of nine genes involved in carotenoid synthesis or degradation [40]. Moreover, carotenoids are very sensitive to heat, oxygen, light, acids, and so on [57]. The detrimental effects on carotenoid compounds in sorghum grain processing should be avoided or reduced.

### 2.3. Vitamin E

α-Tocopherol, β-tocopherol, γ-tocopherol, and δ-tocopherol are the most studied tocochromanols in sorghum (Table 3). Cardoso et al. (2015) reported that γ-tocopherol was the major tocochromanol in sorghum, followed by a-tocopherol, and the vitamin E contents (280.7–2962.4 μg/100 g in wet basis) in sorghum varied significantly [58]. Chung et al. (2013) showed that β-tocopherol was the major tocopherol in sorghum and the vitamin E content in sorghum grains differs with different genotypes [42]. Therefore, the total content and profile of vitamin E in sorghum varies significantly. Moreover, the farming environment or location significantly affect the vitamin E profile and levels in sorghum grains [42].

### 2.4. Amines

Amines are a class of low-molecular-mass nitrogenous bases and can be divided into biogenic amines and polyamines. Paiva et al. (2015) first reported the composition and content of bioactive amines in different sorghum lines. The study showed that spermine and spermidine were the prevalent amines, followed by putrescine and cadaverine, and that the polyamines represented 60–100% of the total amines [43]. Therefore, sorghum is a main source of polyamines. 

### 2.5. Policosanols and Phytosterols

Phytosterols are plant-originated steroids. β-Sitosterol, campesterol, and stigmasterol have been isolated, and β-sitosterol was found to be the main phytosterol in sorghum grains (Table 3). The sorghum genotype, cultivation location, and extraction process can affect the contents of phytosterols [42,44,45]. 

Policosanols are a class of aliphatic alcohols of high molecular weight and have various bioactivities [59]. C26 policosanol, C28 policosanol, C30 policosanol, and C32 policosanol were isolated and detected, and C28 policosanol was found to be the main policosanol in sorghum (Table 3). The determination method can affect the contents of phytosterols [45]; the extraction and detection methods of phytosterols need to be studied in future.

## 3. Biological Activities of Sorghum Grains

The study of biological activities of sorghum has risen considerably in recent years. Here, we summarize the multifarious health-promoting properties of sorghum reported in the literature and pay special attention to the potential mechanisms and related active compounds.

### 3.1. Antioxidative Property

The evaluation of the antioxidative property should be performed using multiple methods based on different mechanisms in order to avoid underestimation [46]. Various methods, such as 2,2-diphenyl-1-picrylhydrazyl (DPPH) assay, 2,2′-azino-bis-3-ethylbenzthiazoline-6-sulphonic acid (ABTS) assay, oxygen radical absorbance capacity (ORAC) assay, and ferric ion reducing antioxidant power (FRAP) assay have been used to measure the antioxidative property of sorghum (Table 4).

Awika et al. (2005) showed that the sorghum genotype significantly affected the antioxidative properties of sorghum grains and bran, as measured by DPPH, ABTS, and ORAC. The brown sorghum grains had the highest antioxidative property due to the presence of tannins, and the black sorghum bran had a higher antioxidative property than the white sorghum bran and red wheat bran due to its high 3-deoxyanthocyanin content [25]. Some studies have shown that total phenolic compound content was largely responsible for the antioxidative property of sorghum [12,26,37]. However, the kind of phenolic compound that contributes a major portion of antioxidant activity varies in sorghum genotypes. While a study showed that various extracts exhibited significant antioxidative property that did not correlate with phenolic content, this was probably because of the antioxidative properties of other bioactive compounds [22]. However, up to now, the relationship between bioactive compounds, except for phenolic compounds and the antioxidative property in sorghum, has not been reported.

Moreover, the red or black sorghum extracts usually show a lower DPPH value than ABTS or ORAC values. This may be because anthocyanins are the major extractable phenolic compounds from red or black sorghums and are the major contributors of the antioxidant activity in sorghum samples. Meanwhile, a similar absorption spectrum of anthocyanins and DPPH causes a color interference with the DPPH chromogen, resulting in a relatively lower DPPH value [8]. 

### 3.2. Anticancer Property

Epidemiological studies and modern pharmacological research have shown the effect of sorghum on the inhibition of cancer. Table 4 shows the sorghum grains and extracts that can inhibit colon cancer, ovarian cancer, lung cancer, benign prostatic hyperplasia, and hepatocellular carcinoma. Phenolic compounds such as 3-DXAs, procyanidin, apigenin, and naringenin have been the main substances to resist the development of cancer [7,13,16,27,28,61,62,64,65].

Various mechanisms can explain the cancer prevention of phytochemicals. STAT3 (signal transducers and activators of transcription (STATs)) is an oncogene that can be activated by several steps, such as phosphorylation, dimerization, and nuclear translocation [61]. Sorghum extracts inhibit STAT3/DNA binding and transcription promoter activity and phosphorylation by inhibiting the expression and phosphorylation of non-ligand activated tyrosine kinase Jak2, and, finally, they inhibit the nuclear export of phosphorylated STAT3 [61]. Moreover, STAT3 and Akt (protein kinase B) molecules can resist apoptosis in cancer cells. Sorghum extracts have suppressed the expression of both STAT3 and Akt with their phosphorylation, and increased the expression of Bax, caspase-3, and the cleavage of caspase-3, and have further increased apoptosis [61]. Nuclear PI3K (phosphatidylinositol 3-kinase) signaling can regulate the antiapoptotic signaling of the nerve growth-factor in different cell types, and sorghum extracts have suppressed these critical anti-apoptotic factors [61]. In brief, sorghum extracts can inhibit both Jak2/STAT3 and PI3K/Akt/mTOR pathways, resulting in the inhibition of the proliferation, cell cycle progression, angiogenesis, migration, invasion, and induction of apoptosis.

In addition, another study showed that the sorghum 3-DXAs mediated apoptosis by stimulating the p^53^ gene and down-regulating the _bcl_ 2 gene in MCF 7 [63]. Furthermore, Yang et al. (2015) reported that sorghum ethyl-acetate extract was effective at activating estrogen receptor -β (ERβ), and ERβ activation can contribute to colon cancer prevention [65]. Furthermore, sorghum ethyl-acetate extract decreased the mRNA expressions of the androgen receptor and 5α-reductase II, and improved the protein-expressed ratio of Bax/Bcl-2 and the oxidative status of benign prostatic hyperplasia induced by testosterone in Sprague–Dawley rats [66].

### 3.3. Antidiabetic Property

Sorghum extracts or products have been effective for diabetic therapy (Table 4). Three study types, in vitro chemistry-based, in vivo animal trial, and in vivo preclinical trial, have been used to study the effects of sorghum extracts or products on diabetes mellitus. In vitro chemistry-based study mainly focusses on the effects of sorghum extracts or products on α-glucosidase and α-amylase, and studies showed that sorghum extracts had strong inhibitory effects on α-glucosidase and α-amylase [12,15,68,72]. In vivo animal trials showed that sorghum extracts or products could significantly protect against hyperglycemia and they suppressed glucose utilization by changing the metabolism of sugar [67,70,71]. Anunciacao et al. (2018) showed that sorghum drink consumption, especially the sorghum 3-DXAs drink, resulted in a lower glycaemic response by in vivo preclinical trial [73]. The main antidiabetic substance in sorghum flavonoids have been condensed tannins and 3-DXAs [12,71,72,73].

### 3.4. Anti-Inflammatory Property

Inflammation is a local response to infection and injury caused by the immune system against external and internal stimuli [23]. Macrophages are recruited to inflammatory sites and lipopolysaccharide (LPS) in macrophages can induce the production of inflammatory cytokines, such as tumor necrosis factor-a (TNF-a), interleukin-1 (IL-1), interleukin-6 (IL-6), interleukin-1β (IL-1β), and interleukin-8 (IL-8), and inflammatory mediators such as nitric oxide (NO) and prostaglandin E2 (PGE2) [8,23,24,76]. NO and PGE2 are synthesized by inducible nitric oxide synthase (iNOS) and cyclooxygenase-2 (COX-2), respectively [23,76]. Multiple in vitro and in vivo studies have shown that sorghum extracts and products can suppress inflammation by reducing the expression of these inflammatory molecules. For example, red sorghum acetone extract significantly suppressed the LPS-induced IL-1β, IL-6, and COX-2 mRNA expressions in RAW 264.7 mouse macrophage cells [8]. Moreover, sorghum 50% methanol (including 2% formic acid) reduced the production of pro-inflammatory cytokines IL-1β and IL-18 in LPS-primed and ATP-activated THP-1 human macrophages by reducing caspase-1 activation and ROS production in THP-1 human macrophages [75]. Extruded sorghum cereal alleviated the inflammation in patients with chronic kidney disease by decreasing the C-reactive protein and malondialdehyde serum levels [60]. Moreover, an in-depth study revealed that caffeoylglycolic acid methyl ester (a major constituent of sorghum) exhibited anti-inflammatory activity via the Nrf2/heme oxygenase-1 pathway. However, the main bioactive compounds studied regarding the anti-inflammatory property were phenolic compounds, and there is less research on which bioactive compounds of sorghum and which mechanisms induce this anti-inflammatory property.

### 3.5. Antiobesity Property

Obesity is characterized by abnormal or excessive fat accumulation. The peroxisome proliferator, a central regulator of adipogenesis, can activate the peroxisome proliferator-activated receptor-γ (PPAR-γ), which coordinates the expression of specific adipogenic genes such as fatty acid synthase (FAS) and lipoprotein lipase (LPL) [53]. Extruded sorghum flour can reduce the percentage of adiposity, fatty acid synthase gene expression, and the adipocyte hypertrophy in obese Wistar rats [53]. Another study showed that extruded sorghum flour reduced hepatic lipogenesis by increasing adiponectin 2 receptor gene expression and the gene and protein expressions of PPARα, and found that the main substances affecting adipogenesis were luteolinidin, apigeninidin, 5-methoxy-luteolinidin, and 7-methoxy-apigeninidin, determined by molecular docking analysis [78]. It is necessary to find other acting substances and verify their role through in vivo and in vitro studies.

## 4. Conclusions and Future Perspectives

As people pay more and more attention to health, sorghum is an increasingly important cereal food with important health-promoting properties. It is an important source of bioactive compounds, such as 3-deoxyanthocyanidins. Many studies have confirmed that sorghum grains and sorghum products have many biological activities, such as anticancer, antidiabetic, anti-inflammatory, and antiobesity properties, from in vitro and in vivo studies.

However, compared with phenolic compounds in sorghum, other bioactive compounds have been researched less. With the development of detection technology, such as mass spectrometry (MS) and nuclear magnetic resonance (NMR), new bioactive compounds may be detected. In addition, as the bioactive compounds vary greatly among different sorghum germplasm resources, the establishment of a sorghum phenolic compounds database is necessary for breeding or industrial use. Furthermore, more attention should be paid to the bound phenolics and their biological activities. Moreover, the biological activities of sorghum need to be further explored, and additional studies elucidating the mechanisms of biological activities of sorghum’s bioactive compounds using clinical trials are necessary.

## Figures and Tables

**Table 1 foods-10-02868-t001:** Total phenolic compounds (TPC) in sorghum grains.

Food Matrix	Seed Coat Color	Extraction Method	TPC (mg GAE/g)	Reference
SSG 59-3 whole grain	Red	1% HCl/methanol (*v/v*) for 2 h with shaking	10.41	[10]
G-46 whole grain	Brown	1% HCl/methanol (*v/v*) for 2 h with shaking	9.56	[10]
PC-5 whole grain	Pearl white	1% HCl/methanol (*v/v*) for 2 h with shaking	1.91	[10]
S-713 whole grain	White	1% HCl/methanol (*v/v*) for 2 h with shaking	1.74	[10]
Cofs29 whole grain	Black	1% HCl/methanol (*v/v*) for 2 h with shaking	8.44	[10]
Terral Rev 9924 whole grain		Ethanol/water/formic acid (50:48:2)	1.21	[16]
Terral Rev 9924 whole grain		Methanol/water/formic acid (50:48:2 *v/v/v*)	0.82	[16]
Pioneer 84P8D whole grain		Ethanol/water/formic acid (50:48:2)	0.89	[16]
Pioneer 84P8D whole grain		Methanol/water/formic acid (50:48:2 *v/v/v*)	0.82	[16]
Dekalb Dk-54-00 whole grain		Ethanol/water/formic acid (50:48:2)	0.86	[16]
Dekalb Dk-54-00 whole grain		Methanol/water/formic acid (50:48:2 *v/v/v*)	0.76	[16]
Ffr353 whole grain		Ethanol/water/formic acid (50:48:2)	0.92	[16]
Ffr353 whole grain		Methanol/water/formic acid (50:48:2 *v/v/v*)	0.84	[16]
Dynagro Dg765B whole grain		Ethanol/water/formic acid (50:48:2)	1.12	[16]
Dynagro Dg765B whole grain		Methanol/water/formic acid (50:48:2 *v/v/v*)	1.07	[16]
Pioneer 83P99 whole grain		Ethanol/water/formic acid (50:48:2)	0.95	[16]
Pioneer 83P99 whole grain		Methanol/water/formic acid (50:48:2 *v/v/v*)	0.86	[16]
Dekalb Dk-51-01whole grain		Ethanol/water/formic acid (50:48:2)	1.07	[16]
Dekalb Dk-51-01whole grain		Methanol/water/formic acid (50:48:2 *v/v/v*)	0.9	[16]
Terral Rev 9782 whole grain		Ethanol/water/formic acid (50:48:2)	1.34	[16]
Terral Rev 9782 whole grain		Methanol/water/formic acid (50:48:2 *v/v/v*)	1.25	[16]
Terral Rev 9562 whole grain		Ethanol/water/formic acid (50:48:2)	1.08	[16]
Terral Rev 9562 whole grain		Methanol/water/formic acid (50:48:2 *v/v/v*)	0.91	[16]
Terral Rev 9562 whole grain		Ethanol/water/formic acid (50:48:2)	0.95	[16]
Terral Rev 9562 whole grain		Methanol/water/formic acid (50:48:2 *v/v/v*)	0.84	[16]
Sorghum whole grain	Red	40% methanol	~1.8	[8]
Sorghum whole grain	Red	60% methanol	~2.1	[8]
Sorghum whole grain	Red	80% methanol	~1.8	[8]
Sorghum whole grain	Red	40% ethanol	~2.3	[8]
Sorghum whole grain	Red	60% ethanol	~2.3	[8]
Sorghum whole grain	Red	80% ethanol	~1.8	[8]
Sorghum whole grain	Red	40% acetone	~2.7	[8]
Sorghum whole grain	Red	60% acetone	~2.6	[8]
Sorghum whole grain	Red	80% acetone	~2.5	[8]
Sorghum whole grain	Red	Acetone/water mixture (60:40, *v/v*), 10:1	~2.3	[8]
Sorghum whole grain	Red	Acetone/water mixture (60:40, *v/v*), 20:1	~2.5	[8]
Sorghum whole grain	Red	Acetone/water mixture (60:40, *v/v*), 30:1	~2.6	[8]
Sorghum whole grain	Red	Acetone/water mixture (60:40, *v/v*), 40:1	~2.55	[8]
Sorghum whole grain	Red	Methanol	47.86	[21]
Sorghum whole grain	White	Methanol	34.78	[21]
Sorghum whole grain	White	Water extraction	0.763	[22]
Sorghum whole grain	White	Methanol extraction	0.461	[22]
Sorghum whole grain	White	Ethanol extraction	0.486	[22]
Sorghum whole grain	White	t-Butanol extraction	0.524	[22]
Sc84Mx whole grain	Black	Water extraction	8.5	[23]
Sc84Mx whole grain	Black	Ethanol extraction	9.58	[23]
Sc84Mx whole grain	Black	0.1% *v/v* HCl extraction	9	[23]
Sc84Mx whole grain	Black	Ethanol with 0.1% *v/v* HCl extraction	18.26	[23]
Sc84Ks whole grain	Black	Water extraction	8.23	[23]
Sc84Ks whole grain	Black	Ethanol extraction	10.24	[23]
Sc84Ks whole grain	Black	0.1% *v/v* HCl extraction	8.5	[23]
Sc84Ks whole grain	Black	Ethanol with 0.1% *v/v* HCl extraction	19.6	[23]
Pi570481 whole grain	Black	Water extraction	1.42	[23]
Pi570481 whole grain	Black	Ethanol extraction	6.02	[23]
Pi570481 whole grain	Black	0.1% *v/v* HCl extraction	3.24	[23]
Pi570481 whole grain	Black	Ethanol with 0.1% *v/v* HCl extraction	12.61	[23]
BRS 309 whole grain	White		6.82	[24]
BRS 305 whole grain	Light brown		0.84	[24]
BRS 310 whole grain	Red		0.95	[24]
Sumac whole grain	Brown	Aqueous acetone (70%)	22.5	[25]
Sc103 whole grain	Brown	Aqueous acetone (70%)	13.5	[25]
Tx430-Cs whole grain	Black	Aqueous acetone (70%)	7.6	[25]
Tx430-V whole grain	Black	Aqueous acetone (70%)	9.8	[25]
ATx631 ×RTx436 whole grain	White	Aqueous acetone (70%)	0.8	[25]
Sorghum Shell	Red	80% ethanol solvent ratio of 1:15 at 50 °C in a 0.32 W cm^−2^ ultrasonic intensity	52.23	[26]
Macia bran	White	50% *v/v* ethanol, shaken for 2 h	~2.5	[27]
Sumac bran	Brown	50% *v/v* ethanol, shaken for 2 h	~28	[27]
Pi152653 bran	Black	50% *v/v* ethanol, shaken for 2 h	~58	[27]
Pi152687 bran	Black	50% *v/v* ethanol, shaken for 2 h	~45	[27]
Pi193073 bran	Black	50% *v/v* ethanol, shaken for 2 h	~50	[27]
Pi329694 bran	Black	50% *v/v* ethanol, shaken for 2 h	~68	[27]
Pi559733 bran	Black	50% *v/v* ethanol, shaken for 2 h	~52	[27]
Pi559855 bran	Black	50% *v/v* ethanol, shaken for 2 h	~24	[27]
Pi568282 bran	Black	50% *v/v* ethanol, shaken for 2 h	~70	[27]
Pi570366 bran	Black	50% *v/v* ethanol, shaken for 2 h	~59	[27]
Pi570481 bran	Black	50% *v/v* ethanol, shaken for 2 h	~74	[27]
Pi570484 bran	Black	50% *v/v* ethanol, shaken for 2 h	~54	[27]
Pi570819 bran	Black	50% *v/v* ethanol, shaken for 2 h	~53	[27]
Pi570889 bran	Black	50% *v/v* ethanol, shaken for 2 h	~57	[27]
Pi570993 bran	Black	50% *v/v* ethanol, shaken for 2 h	~53	[27]
Sorghum bran		Subcritical water extraction	42.453	[28]
Sorghum bran		Hot water extraction	31.813	[28]
Sorghum bran	Red	Acetone	0.14	[29]
Sorghum bran	Red	Methanol	0.58	[29]
Sorghum bran	Red	Acidified methanol	0.93	[29]
Sumac sorghum bran	Red	50% ethanol	62.5	[30]
Black sorghum bran	Black	50% ethanol	23.4	[30]
Mycogen sorghum bran	Bronze	50% ethanol	7	[30]
White sorghum bran	White	50% ethanol	3.1	[30]

**Table 2 foods-10-02868-t002:** Phenolic compounds in sorghum grains.

Phenolic Compounds		Content (ug/g)	Source	Ref.
Phenolic acids				
Hydrocinnamic acids	Caffeic acid	13.55–20.80	3 white sorghum varieties	[31]
		1.91	Sorghum grains	[32]
		Soluble 0–523.02; Bound 1.32–161.11	6 red sorghum varieties	[33]
		10.2	White sorghum flour	[34]
		Soluble 5.44; Bound 52.58	Sorghum grain flour	[35]
		No data	8 brown sorghum genotypes	[12]
		19, 11.5	1 red sorghum and 1 white sorghum	[36]
		1.43–3.87	5 sorghum varieties	[10]
	p-Coumaric acid	41.88–71.88	3 white sorghum varieties	[31]
		3.77	Sorghum grains	[32]
		Soluble 90.71–172.44; Bound 193.25–489.18	6 red sorghum varieties	[33]
		4.87	White sorghum flour	[34]
		Soluble 1.47; Bound 81.93	Sorghum grain flour	[35]
		71, 149	1 red sorghum and 1 white sorghum	[36]
		0.68–2.96	5 sorghum varieties	[10]
	Ferulic acid	120.47–163.91	3 white sorghum varieties	[31]
		15.65	Commercial sorghum grains	[37]
		6.25	Sorghum grains	[32]
		Soluble 291.99–743.65; Bound 949.46–2210.92	6 red sorghum varieties	[33]
		13.4	White sorghum flour	[34]
		Soluble 2.76; Bound 420.96	Sorghum grain flour	[35]
		91.5, 293	1 red sorghum and 1 white sorghum	[36]
		0.81–2.86	5 sorghum varieties	[10]
	Sinapic acid	8.22	Sorghum grains	[32]
		10.5, 17.5	1 red sorghum and 1 white sorghum	[36]
	Chlorogenic acid	235.91–293.19	2 sorghum varieties	[21]
		Soluble 2.95; Bound 9.78	Sorghum grain flour	[35]
		11.5, 25	1 red sorghum and 1 white sorghum	[36]
	Cinnamic acid	9.76–15.02	3 white sorghum varieties	[31]
		0, 11.5	1 red sorghum and 1 white sorghum	[36]
Hydrobenzoic acids	Protocatechuic acid	150.28–178.22	3 white sorghum varieties	[31]
		3.59	Sorghum grains	[32]
		6.18	White sorghum flour	[34]
		Soluble 3.92; Bound 43.61	Sorghum grain flour	[35]
		83.5, 142.5	1 red sorghum and 1 white sorghum	[36]
		1.31–5.88	5 sorghum varieties	[10]
	*p*-Hydroxybenzoic acid	6.13–16.39	3 white sorghum varieties	[31]
		13.3	White sorghum flour	[34]
		19, 11.5	1 red sorghum and 1 white sorghum	[36]
	3,4-Dihydroxybenzoic acid	Soluble 0–369.52; Bound 33–454.54	6 red sorghum varieties	[33]
	Vanillic acid	15.45–23.43	3 white sorghum varieties	[31]
		Soluble 5.81; Bound 14.18	Sorghum grain flour	[35]
		23, 0	1 red sorghum and 1 white sorghum	[36]
	Salicylic acid	63.4	Sorghum grains	[32]
		22.8	White sorghum flour	[34]
	Gallic acid	14.84–21.51	3 white sorghum varieties	[31]
		45.8	Sorghum grains	[32]
		533.10–1005.23	2 sorghum varieties	[21]
		15.65	Commercial sorghum grains	[37]
		Soluble 5.04; Bound 27.98	Sorghum grain flour	[35]
		59, 16.5	1 red sorghum and 1 white sorghum	[36]
	Syringic acid	15.71–17.48	3 white sorghum varieties	[31]
		15.6	Sorghum grains	[32]
		5.5, 25	1 red sorghum and 1 white sorghum	[36]
Flavonoids				
3-Deoxyanthocyanidin	Luteolinidin	Soluble 20.39–57.14; Bound 0.06–0.15	6 red sorghum varieties	[33]
		0.16–0.33	3 sorghum genotypes flours	[24]
		3.16	Sorghum grain flour	[35]
		No data	8 brown sorghum genotypes	[12]
		0.57–1.28	5 sorghum varieties	[10]
	Apigeninidin	Soluble 4.76–13.04; Bound 0.01–0.04	6 red sorghum varieties	[33]
		0.56–1.47	3 sorghum genotypes flours	[24]
		3.17	Sorghum grain flour	[35]
		No data	8 brown sorghum genotypes	[12]
		0.87–3.74	5 sorghum varieties	[10]
	5-Methoxyluteolinidin	Soluble 2.23–6.04; Bound 0.-0.04	6 red sorghum varieties	[33]
		2.04	Sorghum grain flour	[35]
	7-Methoxyapigeninidin	Soluble 5.25–16.82; Bound 0.01–0.05	6 red sorghum varieties	[33]
		0.81	Sorghum grain flour	[35]
	5-Methoxyluteolinidin 7-glucoside	0.18	Sorghum grain flour	[35]
	Luteolinidin 5-glucoside	0.11	Sorghum grain flour	[35]
	7-Methoxyapigeninidin 5-glucoside	0.23	Sorghum grain flour	[35]
	Apigeninidin 5-glucoside	0.07	Sorghum grain flour	[35]
	Luteolinidin anthocyanin	0.09	Sorghum grain flour	[35]
Flavones	Luteolin	112.56–210.70	3 white sorghum varieties	[31]
		No data	8 brown sorghum genotypes	[12]
		1.34, 3.95	1 red sorghum and 1 white sorghum	[36]
		0.68–1.85	5 sorghum varieties	[10]
	Apigenin	25.74–65.58	3 white sorghum varieties	[31]
		2220	Sorghum bran subcritical water extraction	[28]
		No data	8 brown sorghum genotypes	[12]
		0.54, 0	1 red sorghum and 1 white sorghum	[36]
		0.38–2.24	5 sorghum varieties	[10]
	Vitexin	0.50, 0.90	1 red sorghum and 1 white sorghum	[36]
	Hispidulin	No data	8 brown sorghum genotypes	[12]
Flavanones	Naringenin	22.85–28.62	3 white sorghum varieties	[31]
		No data	8 brown sorghum genotypes	[12]
		0.58, 1.11	1 red sorghum and 1 white sorghum	[36]
		0.36–1.16	5 sorghum varieties	[10]
	Naringenin hexoside	13,330	Sorghum bran subcritical water extraction	[28]
	Eriodictyol	No data	8 brown sorghum genotypes	[12]
Flavonols	Kaempferol	17.88–36.44	3 white sorghum varieties	[31]
		No data	8 brown sorghum genotypes	[12]
		0.33, 0.43	1 red sorghum and 1 white sorghum	[36]
	Quercetin	22.34–29.43	3 white sorghum varieties	[31]
		560.28–613.82	2 sorghum varieties	[21]
		21.43	Commercial sorghum grains	[37]
		0.17, 0.49	1 red sorghum and 1 white sorghum	[36]
	Quercetin diglucoside	8420	Sorghum bran subcritical water extraction	[28]
	Rutin	10,290	Sorghum bran subcritical water extraction	[28]
		0.42, 1.61	1 red sorghum and 1 white sorghum	[36]
Flavanols	Catechin	5.58–6.13	3 white sorghum varieties	[31]
		194.15–534.88	2 sorghum varieties	[21]
		5.58	Commercial sorghum grains	[37]
		No data	8 brown sorghum genotypes	[12]
		3.61, 4.57	1 red sorghum and 1 white sorghum	[36]
	Epicatechin	112,860	Sorghum bran subcritical water extraction	[28]
Dihydroflavonol	Taxifolin	27,020	Sorghum bran subcritical water extraction	[28]
		No data	8 brown sorghum genotypes	[12]
		11.95–34.96	5 sorghum varieties	[10]
	Taxifolin hexoside I	25,470	Sorghum bran subcritical water extraction	[28]
	Taxifolin hexoside II	3680	Sorghum bran subcritical water extraction	[28]
Anthocyanins	Cyanidin	No data	8 brown sorghum genotypes	[12]
Isoflavones	Glycitein	No data	8 brown sorghum genotypes	[12]
	Formononetin	No data	8 brown sorghum genotypes	[12]
	Ononin	No data	8 brown sorghum genotypes	[12]
Stilbenoids				
	*trans*-Resveratrol	No data		[38]
	*trans*-Piceid	No data		[38]
Tannins				
	Dimer procyanidin	178,860	Sorghum bran subcritical water extraction	[28]
	Trimer procyanidin	51,380	Sorghum bran subcritical water extraction	[28]
	Tetramer procyanidin	167,550	Sorghum bran subcritical water extraction	[28]

**Table 3 foods-10-02868-t003:** Carotenoids, vitamin E, amines, and phytosterols in sorghum grains.

Bioactive Components	Source	Content	Unit	Reference
Carotenoids				
Lutein	Eight sorghum cultivars	0.003–0.174	mg/kg	[39]
	Red and white sorghum cultivars	24.6, 122.3	mg/kg	[36]
	One hundred sorghum genotypes	0.44–63.37	μg/100g	[40]
Zeaxanthin	Eight sorghum cultivars	0.007–0.142	mg/kg	[39]
	Red and white sorghum cultivars	25.3, 73	mg/kg	[36]
	One hundred sorghum genotypes	1.44–58.85	μg/100g	[40]
β-Carotene	Eight sorghum cultivars	0–0.010	mg/kg	[39]
	Five sorghum cultivars	0.54–1.34	ug/g	[10]
	Red and white sorghum cultivars	27, 34.3	mg/kg	[36]
	Three white sorghum cultivars	0.54–1.19	mg/kg	[31]
Vitamin E				
α-Tocopherol	One hundred sorghum genotypes	0–1231.6	μg/100g	[40]
	Five sorghum cultivars	1.22–5.26	µg/g	[10]
	Sorghum flour and seed	0.0846, 0.01247	mg/100 g	[41]
	5 sorghum varieties cultivated in Wonju	41.61–44.99	mg/kg	[42]
	5 sorghum varieties cultivated in Miryang	41.75–47.53	mg/kg	[42]
β-Tocopherol	One hundred sorghum genotypes	0–784.7	μg/100g	[40]
	5 sorghum varieties cultivated in Wonju	63.89–76.87	mg/kg	[42]
	5 sorghum varieties cultivated in Miryang	82.56–112.52	mg/kg	[42]
γ-Tocopherol	One hundred sorghum genotypes	174.6–2109	μg/100g	[40]
	Sorghum flour and seed	0.2008, 0.2244	mg/100 g	[41]
	5 sorghum varieties cultivated in Wonju	32.77–43.11	mg/kg	[42]
	5 sorghum varieties cultivated in Miryang	35.06–51.28	mg/kg	[42]
δ-Tocopherol	One hundred sorghum genotypes	0–379.8	μg/100g	[40]
	5 sorghum varieties cultivated in Wonju	33.37–36.95	mg/kg	[42]
	5 sorghum varieties cultivated in Miryang	31.34–37.98	mg/kg	[42]
α-Tocotrienol	One hundred sorghum genotypes	0–311.9	μg/100g	[40]
β-Tocotrienol	One hundred sorghum genotypes	0–850.5	μg/100g	[40]
γ-Tocotrienol	One hundred sorghum genotypes	0–270.5	μg/100g	[40]
δ-Tocotrienol	One hundred sorghum genotypes	0–484.2	μg/100g	[40]
Amines				
Spermidine	22 lines of sorghum	0.5–18.7	mg/kg	[43]
Spermine	22 lines of sorghum	2.7–27.2	mg/kg	[43]
Putrescine	22 lines of sorghum	0.7–7.2	mg/kg	[43]
Cadaverine	22 lines of sorghum	0–0.6	mg/kg	[43]
Policosanols and phytosterols				
β-Sitosterol	5 sorghum varieties cultivated in Wonju	17.75–32.32	mg/kg	[42]
	5 sorghum varieties cultivated in Miryang	0.37–11.37	mg/kg	[42]
	Dry distiller’s grain lipids	4.1	mg/g	[44]
	Soxtec extraction of whole grain sorghum	1.92	mg/g of lipids	[45]
	Reflux extraction of whole grain sorghum	0.93	mg/g of lipids	[45]
Campesterol	Soxtec extraction of whole grain sorghum	1.04	mg/g of lipids	[45]
	Reflux extraction of whole grain sorghum	0.97	mg/g of lipids	[45]
	Dry distiller’s grain lipids	1.7	mg/g	[44]
Stigmasterol	Soxtec extraction of whole grain sorghum	1.02	mg/g of lipids	[45]
	Reflux extraction of whole grain sorghum	1.08	mg/g of lipids	[45]
	Dry distiller’s grain lipids	4.2	mg/g	[44]
C26 policosanol	Soxtec extraction of whole grain sorghum	1.53	mg/g of lipids	[45]
	Reflux extraction of whole grain sorghum	4.62	mg/g of lipids	[45]
C28 policosanol	Soxtec extraction of whole grain sorghum	2.7	mg/g of lipids	[45]
	Reflux extraction of whole grain sorghum	9.69	mg/g of lipids	[45]
C30 policosanol	Soxtec extraction of whole grain sorghum	1.31	mg/g of lipids	[45]
	Reflux extraction of whole grain sorghum	3.99	mg/g of lipids	[45]
C32 policosanol	Soxtec extraction of whole grain sorghum	0.25	mg/g of lipids	[45]
	Reflux extraction of whole grain sorghum	0.52	mg/g of lipids	[46]

**Table 4 foods-10-02868-t004:** Potential health benefits of sorghum grains.

Potential Health Benefits	Sorghum Substrate	Study Type and Method	Main Results	Reference
Antioxidative property	Sorghum bran aqueous acetone (70%) extracts	In vitro chemistry-based; DPPH, ABTS, ORAC	DPPH: 6.2–202 μmol TE/g; ABTS: 9.8–240 μmol TE/g; ORAC: 6.2–202 μmol TE/g	[25]
	Sorghum shell 80% ethanol extract	In vitro chemistry-based; FRAP, ABTS	FRAP: 77.01 μmol Fe/g; ABTS: 53.22 μmol TE/g	[26]
	Sorghum 200 proof methanol extract	In vitro chemistry-based; DPPH	DPPH: 133.5 μmol TE/100 g	[34]
	Sorghum 70% methanol extract	In vitro chemistry-based; DPPH, FRAP, ORAC	DPPH: 83.76%; FRAP: 0.029 mmol FE/gDM; ORAC: 25.38 μmol TE/g	[37]
	Red sorghum acetone extract	In vitro chemistry-based; DPPH, FRAP, ORAC	DPPH: 1.97 mg Trolox/g; FRAP: 13.71 mg Trolox/g; ORAC: 40.59 mg Trolox/g	[8]
	Sorghum 70% ethanol extract	In vitro chemistry-based; DPPH, ABTS	DPPH IC_50(ug/mL)_: ~90–~360; ABTS IC50_(ug/mL)_: ~200–~360	[12]
	Sorghum water extract, methanol extract, ethanol extract, t-butanol extract	In vitro chemistry-based; DPPH	DPPH IC50(ug/mL): 17.11–18.02	[22]
	Sorghum flour	In vivo animal trial; at 53 days of age, 50 male Rattus norvegicus Wistar rats	Increased levels of enzymes SOD	[24]
	Extruded sorghum cereal	In vivo preclinical trial; patients with chronic kidney disease	Decreased malondialdehyde levels, increased total antioxidant capacity and the enzymatic activity of dismutase	[60]
Anticancer property	Sorghum methanol extract	In vitro cell culture-based; HCT-116 and HCT-15 human colon cancer cells and COS-7, monkey kidney cells	Inhibited the proliferation of human colon cancer cells by inducing G1 phase arrest and apoptosis. Suppressed the Jak2/STAT3 and PI3K/AKT/mTOR pathways	[61]
	Sorghum ethanol extract	In vitro cell culture-based; A27801AP OVCA cells and its paclitaxel-resistant variant A27801AP-X10 (PTX10)	Reduced the proliferation 35 and colony formation of OVCA cells	[16]
	Sorghum 70% ethanol (including 5% citric acid) extract	In vitro cell culture-based; human colon cancer cell lines (HCT15, SW480, HCT116, and HT-29) and noncolon cancer cell lines (3T3-L1, RAW264.7, and HUVEC)	Inhibited the cell proliferation, cell migration and invasion, and induced apoptosis	[7]
	Sorghum methanol extract	In vitro cell culture-based; human leukemia HL-60 and hepatoma HepG2 cell lines	Reduced the viability of HL-60 and HepG2 cells by 90 and 50%	[62]
	Sorghum methanol (including 1% hydrochloric) extract	In vitro cell culture-based; MCF-7 (human breast cancer cell line)	Showed 84.09% of inhibition in the proliferation of MCF 7 cells by stimulation of P^53^ gene and down-regulation of Bcl-2 gene	[63]
	Sorghum 70% ethanol extract	In vivo animal trial; fifty malemice (C57BL/6J) aged 46 weeks, weighing 18 (2 g)	Inhibited tumor growth and metastasis formation by suppressing vascular endothelial growth factor (VEGF) production	[13]
	Sorghum 70% aqueous acetone (acidified with 0.1% HCl) extract	In vitro cell culture-based; murine hepatoma Hepa 1c1c7 and human colon carcinoma HT-29 cell lines	Had strong antiproliferative activity against HT-29 cells	[64]
	Sorghum 70% (*v/v*) aqueous acetone extracts	In vitro cell culture-based; young adult mouse colonocytes (YAMC) cells	Apigenin and naringenin reduced ER-mediated YAMC cell growth	[65]
	Sorghum bran subcritical water extraction	In vitro cell culture-based; HepG2 cells	There was a remarkable increase in inhibition effect on HepG2 cells after exposed to the sorghum bran extracts	[28]
	Donganme sorghumethyl-acetate extract (DSEE)	In vivo animal trial; male Sprague–Dawley (SD) rats aged 7 weeks	Inhibited weight gain of the prostate; decreased mRNA expressions of androgen receptor and 5α-reductase II; and improved histopathological symptoms, the protein-expressed ratio of Bax/Bcl-2, and the oxidative status of BPH induced by testosterone in SD rats	[66]
	Sorghum 50% ethanol extract	In vitro cell culture-based; human hepatocellular carcinoma (HepG2) and colorectal adenocarcinoma (Caco2) cells	Reduced cell viability by inducing apoptosis and cell cycle arrest following production of reactive oxygen species and oxidative DNA damage	[27]
Antidiabetic property	Ethanolic extracts from sorghum	In vivo animal trial; six-week-old male Wistar rats	Reduced the concentration of triglycerides, total and LDL-cholesterol and glucose by inhibition of hepatic gluconeogenesis	[67]
	Sorghum 70% ethanol extract	In vitro chemistry-based; inhibitory activity of α-glucosidase and α-amylase	Strongly inhibited degradation of starch by α-glucosidase as well as porcine pancreatic and human salivary α-amylases	[68]
	Sorghum 70% ethanol extract	In vitro chemistry-based; inhibitory activity of α-glucosidase and α-amylase	SOR 11, SOR 17, and SOR 33 exhibited significantly higher percentage inhibitory activity of α-glucosidase and α-amylase, showed significantly potent inhibition of AGEs formation with IC50 values	[12]
	Sorghum lipid extract	In vivo animal trial; male F1B Syrian hamsters aged 7 wk and weighing 80 g	Increased cholesterol excretion and decreased plasma and liver cholesterol concentration in hamsters	[69]
	Fermented sorghum	In vivo animal trial; healthy female Wistar albino rats weighing 150–200 g	Statistically significant decrease in liver dysfunction indices and markers of oxidative damage. Significantly decreased the relative expression of superoxide dismutase, glutathione peroxidase, glucokinase, hosphofructokinase, and hexokinase genes	[70]
	Kafirin microparticle encapsulated sorghum, condensed tannins	In vivo animal trial; healthy, adult (15 week) male Sprague Dawley rats (260–350 g)	SCT-KEMS prevented a blood glucose spike and decreased the maximum blood glucose level by 11.8%	[71]
	Alcoholic extraction of SCT from sorghum bran	In vitro chemistry-based; inhibitory activity of α-glucosidase and α-amylase	Retained their inhibitory activity against y α-glucosidase and α-amylase	[72]
	Sorghum drinks	In vivo preclinical trial; volunteers	Reduced the glycaemic curve	[73]
Anti-inflammatory property	50% ethanol extracts from sorghum bran	In vivo animal trial; male Swiss Webster mice weighing 20–24 g	Significantly inhibited the secretion of the pro-inflammatory cytokines interleukin-1b and tumor necrosis factor-a	[30]
	Sorghum 95% ethanol extract	In vitro chemistry-based; inhibitory activity of α-glucosidase	Had strong inhibitory effects on blood coagulation, aglucosidase enzyme	[15]
	Caffeoylglycolic acid methyl ester (CGME) and 1-ocaffeoylglycerol	In vitro cell culture-based; RAW264.7 cells, C57BL/6 mice	Induced HO-1 protein and mRNA expression. Increased nuclear translocation of nuclear factor-E2-related factor 2 (Nrf2) and knockdown of Nrf2 by siRNA blocked CGME-mediated HO-1 induction	[74]
	Sorghum 50% methanol (including 2% formic acid)	In vitro cell culture-based; THP-1 human macrophages	Reduced the production of proinflammatory cytokines IL-1β and IL-18 in LPS-primed and ATP-activated THP-1 human macrophages by reducing caspase-1 activation and ROS production	[75]
	Sorghum flour	In vivo animal trial; at 53 days of age, 50 male Rattus norvegicus Wistar rats	Reduced the production of IL-8, TNF-α, and IL-10	[24]
	Sorghum 95% EtOH extract	In vitro cell culture-based; RAW264.7 macrophages	Potential inhibitory effects against LPS-induced NO production in macrophage RAW264.7 cells	[76]
	Sorghum water and ethanol extracts	In vitro cell culture-based; RAW 264.7 macrophages	Inhibited the production of NO, interleukin-6 (IL-6)	[23]
	Red sorghum acetone extract	In vitro cell culture-based; RAW 264.7 mouse macrophage cells	Significantly suppressed the LPS-induced IL-1β, IL-6, and COX-2 mRNA expressions	[8]
	White sorghum aqueous acetone (70%, *v/v*) extracts	In vitro cell culture-based; nonmalignant colon myofibroblast CCD18Co cell	Significantly reduced proinflammatory cytokines (TNF-α, IL-6, IL-8) mRNA and protein expression	[77]
	Extruded sorghum flour	In vivo animal trial; male Wistar rats, aged 21 days and weighing 69 ± 5 g	Inhibited the secretion of IL-1β, TNF-α, and nitric oxide	[53]
	Extruded sorghum cereal	In vivo preclinical trial; patients with chronic kidney disease	Alleviated the inflammation in patients with chronic kidney disease by decreasing the C-reactive protein and malondialdehyde serum levels	[60]
Antiobesity property	Extruded sorghum flour	In vivo animal trial; male Wistar rats, aged 21 days and weighing 69 ± 5 g	Reduced fatty acid synthase gene expression, TNF-α, blood levels of glucose, and the adipocyte hypertrophy	[53]
	Extruded sorghum flour	In vivo animal trial; Wistar rats (Rattus novergicus) adult males (60 days old)	Reduced the body mass index and liver weight, reduced hepatic lipogenesis by increasing adiponectin 2 receptor gene expression and gene and protein expressions of peroxisome proliferator-activated receptor α	[78]
	Red sorghum Flaked biscuits	In vivo preclinical trial; 46 females and 14 males	Weight lost	[79]

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
