# Peer review of "Bioactive Compounds and Biological Activities of Sorghum Grains"

_foods, 2021, doi:10.3390/foods10112868_

Round 1

Reviewer 1 Report

The manuscript “Bioactive compounds and biological activities, and processing of sorghum grains" summarizes the biological activities, phytochemical constituents, and different types of sorghum grains processing. I thank the authors for their efforts in his review. The following reasons:

1-  The manuscript is a listing-type review that did not discuss the topic critically and did not provide the readers critical opinions depending on the reviewed literature. 

2-  Recent reviews have comprehensively described the biological effects of sorghum grains, for example, I- Xiong, Y., Zhang, P., Warner, R. D., & Fang, Z. (2019). Sorghum grain: From genotype, nutrition, and phenolic profile to its health benefits and food applications. Comprehensive reviews in food science and food safety, 18(6), 2025-2046.‏; II- Shahidi, F., Danielski, R., & Ikeda, C. (2021). Phenolic compounds in cereal grains and effects of processing on their composition and bioactivities: A review. Journal of Food Bioactives, 15.‏; III- ZHoNGxIANG, F. A. N. G. (2018). Sorghum non-extractable polyphenols: Chemistry, extraction and bioactivity. Non-extractable Polyphenols and Carotenoids: Importance in Human Nutrition and Health, 326.‏; Ed Nignpense, B., Francis, N., Blanchard, C., & Santhakumar, A. B. (2021)., IV- Bioaccessibility and Bioactivity of Cereal Polyphenols: A Review. Foods, 10(7), 1595.‏

Some of these recently published reviews even provided more valuable comparisons between different types of grains. Accordingly, what is the new contribution and/or discussion the authors provided in this manuscript.

3-  The manuscript needs extensive English corrections and should be revised by a native English speaker.

Minor Issues to be considered before further processing:

  1. Please rephrase the abstract. I found that its statements were just copied and pasted from other parts of the review.
  2. Lines 32 and 33: The statement is not accurate and needs to be rephrased with a moderate tone.
  3. Line 47: Please replace And with additionally or moreover in the beginning of new statement through all the manuscript.
  4. Table 1: p-coumaric acid should be corrected to be “p-coumaric acid”. Please revise and correct throughout the whole manuscript.

Author Response

Dear Reviewer,

I greatly appreciate both your help and that of the referees concerning improvement to this paper. I hope that the revised manuscript is now suitable for publication. The revised parts can be seen in the revised state. Here below is our description on revision according to the reviewers’ comments.

  1. We have rewritten the abstract.
  2. Replaced And with additionally or moreover in the beginning of new statement through all the manuscript.

    In addition, we added a table “Total phenolic compounds (TPC) in sorghum grains” and related statements, deleted the part of “Effcets of processing on bioactive components and bioactivities of sorghum grains”. We revised the English grammar and style through the manuscript.

Reviewer 2 Report

In the section "introduction" - it would be necessary to add information on the countries where sorghum is grown, where active research of this species is carried out as food, medicinal, where new varieties are created and what direction in breeding. Cards could be added: 1) - the growth of wild sorghum; 2) - countries that currently cultivate sorghum. These cards, among other things, could be used as a graphic abstraction.

It would be nice to add information about centers for the study of the most important crops on the planet.

It is somewhat broader to present information on the main cereals - wheat, rice, corn and barley as sources of not only the basis of human and animal nutrition, but also important biologically active compounds.

My comments are more of a recommendatory nature, which are aimed at improving the presentation of the author's material.

Author Response

Dear Reviewer,

I greatly appreciate both your help and that of the referees concerning improvement to this paper. I hope that the revised manuscript is now suitable for publication. The revised parts can be seen in the revised state. Here below is our description on revision according to the reviewers’ comments.

We added information on the countries where sorghum is grown. In addition, we added a table “Total phenolic compounds (TPC) in sorghum grains” and related statements, deleted the part of “Effcets of processing on bioactive components and bioactivities of sorghum grains”. We revised the English grammar and style through the manuscript.

Reviewer 3 Report

This review paper gives supportive information of the reasons why sorghum grains are important in nutrition and how different types of processing have impact on biological activities of sorghum grains. Thus, the manuscript should be accepted for publication. I would have just some minor comments/suggestions for the authors:

in vivo and in vitro must be italic. Check text and tables and correct.  

Table 1 and 2 should be the same. Add source, content and unit columns in table 1.

Line 245, 246 The sentence is not complete

Line 264-8 Add reference.

Author Response

Dear Reviewer,

I greatly appreciate both your help and that of the referees concerning improvement to this paper. I hope that the revised manuscript is now suitable for publication. The revised parts can be seen in the revised state. Here below is our description on revision according to the reviewers’ comments.

We added source, content and unit columns in table 2 and turned in vivo and in vitro into italics.

In addition, we added a table “Total phenolic compounds (TPC) in sorghum grains” and related statements, deleted the part of “Effcets of processing on bioactive components and bioactivities of sorghum grains”. We revised the English grammar and style through the manuscript.

Reviewer 4 Report

Since sorghum is ranked as the fifth most commonly used cereal worldwide and contains various bioactive components with extensive biological activities, a review paper on this topic is a good idea but the paper is not well organized, it is too long and difficult to read. The title is not adequate and Tables 1 and 2 are not consistently presented. A column with Source of Phenolic compounds and their content should also be added in Table 1. A different concept of review of a particular group of bioactive components has been applied, so a significantly smaller review is devoted to the biological activity of typical cereals phenolic components. I think that the processing impact could / should be a special reviewed paper because in this volume at the end of the paper it acts as a surplus. Many references, perhaps subsequently added, are not adequately cited (without square brackets) in the text.

Author Response

Dear Reviewer,

I greatly appreciate both your help and that of the referees concerning improvement to this paper. I hope that the revised manuscript is now suitable for publication. The revised parts can be seen in the revised state. Here below is our description on revision according to the reviewers’ comments.

We added source, content and unit columns in table 2, added a table “Total phenolic compounds (TPC) in sorghum grains” and related statements, deleted the part of “Effcets of processing on bioactive components and bioactivities of sorghum grains”. In addition, we revised the English grammar and style through the manuscript.

Round 2

Reviewer 1 Report

The manuscript has been improved. However, I think this manuscript is a listing-type manuscript that does not provide a critical overview of the reviewed topic.

Author Response

Dear Reviewer,

Thank you very much for your advice.

Reviewer 4 Report

The manuscript has been sufficiently improved to warrant publication in Foods.

Best regards

Author Response

(The authors gave the same response as above.)
